# Mutation of *OsLPR3* Enhances Tolerance to Phosphate Starvation in Rice

**DOI:** 10.3390/ijms24032437

**Published:** 2023-01-26

**Authors:** Hao Ai, Xiuli Liu, Zhi Hu, Yue Cao, Nannan Kong, Feiyan Gao, Siwen Hu, Xing Shen, Xianzhong Huang, Guohua Xu, Shubin Sun

**Affiliations:** 1Center for Crop Biotechnology, College of Agriculture, Anhui Science and Technology University, Fengyang 233100, China; 2State Key Laboratory of Crop Genetics and Germplasm Enhancement, Key Laboratory of Plant Nutrition and Fertilization in Low-Middle Reaches of the Yangtze River, Ministry of Agriculture, Nanjing Agricultural University, Nanjing 210095, China

**Keywords:** *OsLPR3*, rice, phosphate, growth and development, homeostasis

## Abstract

*Low Phosphate Root* (*LPR*) encodes a protein localized to the endoplasmic reticulum (ER) and cell wall. This gene plays a key role in responding to phosphate (Pi) deprivation, especially in remodeling the root system architecture (RSA). An identification and expression analysis of the *OsLPR* family in rice (*Oryza sativa*) has been previously reported, and *OsLPR5*, functioning in Pi uptake and translocation, is required for the normal growth and development of rice. However, the role of *OsLPR3*, one of the five members of this family in rice, in response to Pi deficiency and/or in the regulation of plant growth and development is unknown. Therefore, in this study, the roles of *OsLPR3* in these processes were investigated, and some functions were found to differ between *OsLPR3* and *OsLPR5*. *OsLPR3* was found to be induced in the leaf blades, leaf sheaths, and roots under Pi deprivation. *OsLPR3* overexpression strongly inhibited the growth and development of the rice but did not affect the Pi homeostasis of the plant. However, *oslpr3* mutants improved RSA and Pi utilization, and they exhibited a higher tolerance to low Pi stress in rice. The agronomic traits of the *oslpr3* mutants, such as 1000-grain weight and seed length, were stimulated under Pi-sufficient conditions, indicating that *OsLPR3* plays roles different from those of *OsLPR5* during plant growth and development, as well as in the maintenance of the Pi status of rice.

## 1. Introduction

Phosphorus (P) is an essential macronutrient for plant growth and development. In addition to being a key constituent of molecules, such as nucleic acids, ATP, and membrane phospholipids, P plays crucial roles in signal transduction, energy transfer, photosynthesis, and respiration [1,2]. Inorganic orthophosphate (Pi), the major form of P absorbed and assimilated by plants, is the least accessible macronutrient in many natural and agricultural ecosystems. Its inaccessibility is mainly due to its poor mobility in soils and/or its propensity to form sparingly soluble salts with oxides or hydroxides of aluminum and iron in acidic soils and with calcium and magnesium in alkaline soils [2,3,4]. Its low availability in soil often limits plant growth and productivity in soils across climatic zones [5].

Plants have evolved several strategies that adapt their physiology to Pi concentration variations. Molecular responses to low Pi availability have been divided into two partially independent signaling pathways known as the local response and the systemic response to Pi starvation [6,7]. Systemic responses depend on the internal P status of the plant and include the upregulation of genes involved in the overall enhancement of Pi uptake and internal P-use efficiency, and they are largely controlled by the master regulator PHR1 (a Myb transcription factor) [8,9]. Local responses depend on the level of Pi available in the external medium [6,10]. including changes in root traits (e.g., the inhibition of primary root growth, enhancement in lateral root density, and an increase in the number and length of root hairs) [11,12], and they are modulated by the external level of Pi that is in contact with the root tip [6,10,13]. These changes in the RSA can enhance the ability of the plant to explore the soil for Pi by increasing the root surface area in the top layers of the soil where Pi tends to accumulate [12,14,15,16,17,18].

In *Arabidopsis thaliana*, the contact of the root tip with a low-Pi medium in the presence of iron is sufficient to inhibit primary root growth in response to low Pi availability [13,19]. This process induces determinate development known as root apical meristem exhaustion, which consists of premature cell differentiation and arrested mitotic activity in the root apical meristem, leading to a short root phenotype [12].

In the last two decades, many *Arabidopsis* mutants with altered sensitivities of primary root growth to Pi deficiency have been identified [12,13,18,20,21,22,23]. Two of these mutants, *low phosphate root 1* and *2* (*atlpr1* and *atlpr2*), have lesions in genes encoding the members of the multicopper oxidase family. They are defective in altering RSA in response to low Pi availability. The *atlpr1* and *atlpr2* double mutants have an additive phenotype on the primary root response to Pi deficiency [13]. Another primary root growth mutant, *phosphate deficiency response 2* (*pdr2*), is hypersensitive to low Pi. Its triggering of the root system response to low Pi availability is more sensitive than that of the wild type (WT) [20]. The PDR2 protein interacts genetically with LPR1 in the endoplasmic reticulum (ER) [24]. A low Pi availability inhibits the negative regulation of LPR1 by PDR2 and induces LPR1 transport to the plasma membrane. The plasma membrane localization triggers Fe and callose deposition in the apoplast. These apoplastic changes impair the movement of SHORT ROOT (SHR) and interfere with symplastic communication, which is responsible for root meristem differentiation [18,19]. In addition, the roles of SENSITIVE TO PROTON RHIZOTOXICITY (STOP1) and its target ALUMINUM ACTIVATED MALATE TRANSPORTER 1 (ALMT1), which releases malate (an organic acid thought to chelate cations), making more Pi available for absorption by root hairs, have been identified. The STOP1–ALMT1 module interacts with an unknown pathway parallel with LPR1–PDR2 to inhibit cell division in the stem cell niche via a similar mechanism involving the accumulation of Fe and callose, peroxidase activity, and cell wall thickening [18,22]. Furthermore, the direct illumination of the root surface with blue light is critical and sufficient for the Pi-deficiency-induced inhibition of primary root growth in Arabidopsis seedlings. Blue light and malate, Fe^2+^, Fe^3+^, H_2_O_2_, low pH, and low Pi are essential for the continued generation of ·OH radicals by this Fe redox cycle. A lack of any of these components would abolish the inhibition of primary root growth under Pi deficiency [25,26]. Recently, an additional function of *Atlpr2* was uncovered: beyond that of altering RSA under low Pi conditions, it was found to encode a cell wall ferroxidase involved in the accumulation of excess iron in the phloem apoplast that leads to the remodeling of root development under NH^4+^ stress, which is irreplaceable by LPR1 [27]. 

Although several genes associated with the embryonic and postembryonic development of rice root have been identified [28,29], the role of the homologs of *LPR1* in rice remains elusive. In our earlier study, we identified five *LPR1* homologs in rice (*OsLPR1–5*); among these, *OsLPR3* and *OsLPR5* revealed strong tissue-specific induction during Pi deficiency [30]. However, it is not known whether *OsLPR3* affects the responses of different root traits of rice seedlings during growth under different Pi regimes and/or whether other morphophysiological and molecular traits govern the maintenance of Pi homeostasis.

In this study, *OsLPR3* was functionally identified by analyzing transgenic overexpression and knockout lines. *OsLPR3* expression was induced by low Pi. *OsLPR3* overexpression inhibited the growth of rice but did not affect its Pi status. In addition, the *OsLPR3* mutation increased the primary root length, number of adventitious roots, and soluble Pi concentration, and it promoted the growth and development of rice during Pi deficiency. The agronomic traits of *oslpr3* mutants, such as 1000-grain weight and seed length, were stimulated under Pi-sufficient conditions, indicating the key roles of *OsLPR3* in the growth and development, as well as in the maintenance of Pi homeostasis, of rice.

## 2. Results

### 2.1. OsLPR3 Was Responsive to Pi Deprivation

To investigate the expression patterns of *OsLPR3* at various nutrient deficiencies, Nipponbare seedlings were initially grown in an IRRI solution for 10 days (10-d-old seedlings). The seedlings were then transferred to a complete nutrient solution (CK) or to a nutrient solution lacking either nitrogen (−N), phosphate (−P), potassium (−K), magnesium (−Mg), or iron (−Fe), and they were grown for 7 d. The relative transcript levels of *OsLPR3* in the leaf blades, leaf sheaths, and root samples of the treated seedlings were assessed via a quantitative polymerase chain reaction (qPCR). In CK, the relative transcript levels of *OsLPR3* were high in the roots and low in the leaf blades and leaf sheaths. Moreover, most nutrient deficiencies, such as −N, −K, −Mg, and −Fe, did not exert a significant impact on the transcript levels of *OsLPR3* compared with CK (Figure 1). The relative transcript levels of *OsLPR3* were induced by Pi deficiency in all three plant tissues. In the leaf blades, the relative transcript levels of *OsLPR3* were 150-fold higher in −P than in CK (Figure 1A); however, in the roots, the relative transcript levels of *OsLPR3* were 13-fold higher in −P than in CK (Figure 1C). Thus, after the induction by −P, the relative transcript levels of *OsLPR3* were only three times more abundant in the roots than in the leaf blades. The relative expression of *OsLPR3* remained much lower in the leaf sheaths than in the roots and leaf blades after the Pi deficiency induction.

### 2.2. OsLPR3 Was Localized to the ER

The ER is the entry point for membrane and secretory proteins into the endomembrane system in all eukaryotic cells [31]. It is the site at which these proteins are folded in order to acquire biological functions at their final destination [32]. The protein localization program Softberry (http://linux1.softberry.com/, accessed on 3 September 2022) suggested the putative ER localization of the OsLPR3 protein. For the subcellular localization of the putative ER-targeted proteins, a red fluorescent protein (mCherry) is a potent in vivo marker applicable in diverse plant species [33]. To provide experimental evidence for the ER localization of *OsLPR3*, a reporter gene encoding a green fluorescent protein (EGFP) driven by a CaMV35S promoter was fused upstream and in frame with the *OsLPR3* open-reading frame (*35S::eGFP::OsLPR3*). This construct was transiently co-expressed with *35S::mCherry::HDEL*, which exhibits characteristic red fluorescence in the ER [34] in the leaf epidermal cells of *Nicotiana benthamiana*. Confocal microscopy was used to demonstrate the co-localization of EGFP::OsLPR3 and mCherry::HDEL to the ER (Figure 2). The co-localization of the two proteins was particularly apparent in the cytoplasmic strands extending across the vacuole (white arrows). The ER localization of OsLPR3 was consistent with the localization of AtLPR1 in *Arabidopsis* [24]. 

### 2.3. OsLPR3 Was Involved in Vegetative Growth and RSA of Rice

The functions of *OsLPR3* were explored using transgenic plants in which *OsLPR3* was overexpressed (Ox) and knocked out. The Ox lines with the strongest upregulated expressions of *OsLPR3* (designated Ox5, Ox10, Ox29, and Ox31 (Appendix A)) were selected for further analyses. Southern blot analyses demonstrated that the Ox transgenic lines all arose from independent transformation events (Appendix A). The knockout mutants of *OsLPR3* (designated *oaslpr3-1*, *oslpr3-2*, and *oslpr3-3*) were confirmed after sequencing the lesion site (Appendix A).

The effects of *OsLPR3*-Ox and the knockout mutants on the vegetative growth of the 10-d-old rice seedlings grown hydroponically for an additional 21 d under +P and −P conditions were determined. The plant height of the Ox plants was shorter than that of the WT and *oslpr3* lines under both +P and −P conditions. The plant height of the *oslpr3* lines did not differ from that of WT under +P conditions (Figure 3C). However, under −P conditions, the plant height of the *oslpr3* plants was higher than that of WT (Figure 3D). The root length of both the Ox and knockout lines did not differ from that of WT under +P conditions (Figure 3E). However, both the Ox and knockout lines had similarly long roots under −P conditions compared with those of WT (Figure 3F). The effects of the *OsLPR3*-Ox and knockout lines on the biomass and root–shoot ratio were also determined. The shoot and root biomasses of the *OsLPR3*-Ox plants were repressed under both +P and −P conditions, but no significant differences were observed between their root–shoot ratio and that of WT (Appendix A). The shoot biomass, root biomass, and root–shoot ratio of the *oslpr3* lines were not significantly different from those of WT under +P conditions (Appendix A). However, under −P conditions, both the shoot biomass and root biomass were increased in the *oslpr3* plants, and they were ~54% and ~24% higher in the transgenic plants than in the WT plants, respectively. In addition, the root–shoot ratio was ~22% lower in the *oslpr3* mutants than in WT (Appendix A).

Furthermore, the *OsLPR3*-Ox lines seemed to have a spindly RSA, whereas the *oslpr3* lines had an RSA thicker than that of WT. These observations were examined further by determining the number of adventitious roots in each mutant line. Compared with WT, the number of adventitious roots in the *OsLPR3*-Ox lines was lower under both +P and −P conditions, whereas the number of adventitious roots in the *oslpr3* lines was not significantly different under +P conditions, but it was higher under −P conditions (Appendix A). Moreover, the lateral root densities in the WT and transgenic lines were not significantly different under either Pi-sufficient or Pi-deficient conditions (Appendix A).

### 2.4. Alteration of OsLPR3 Expression Affected the Agronomic Traits of Rice during the Reproductive Stage

The growth inhibition of the *OsLPR3*-Ox lines observed at the seedling stage (Figure 3) was also clear in the 20-week-old mature plants (Figure 4). Both plant height and yield were significantly reduced by *OsLPR3* overexpression (Figure 4A,B). The plant height was also lower in the 20-week-old *oslpr3* plants, although the yield was not significantly affected (Figure 4C). We measured the total tiller number and effective tiller number of the WT and *OsLPR3* transgenic lines. The alteration of *OsLPR3* expression did not affect the total tiller number and effective tiller number of the rice. However, the knockout of *OsLPR3* decreased the seed setting rate of the rice (Appendix A). *OsLPR3* overexpression decreased the 1000-grain weight by 10%, whereas the knockout of *OsLPR3* increased the 1000-grain weight by 5–11% (Figure 4D). Furthermore, the seed width was reduced in the *OsLPR3*-Ox lines but was unchanged in the *oslpr3* knockout lines compared with that of WT (Figure 5A). In contrast, the seed length was significantly greater in the *oslpr3* mutants but was unchanged in the *OsLPR3*-Ox lines compared with that of WT (Figure 5B). These results suggest that *OsLPR3* affects the agronomic traits of rice by regulating seed size.

### 2.5. Alteration of OsLPR3 Expression Affected the Pi Status

The effects of the *OsLPR3*-Ox and knockout mutants on Pi uptake and utilization were determined in the 10-d-old seedlings of the WT and transgenic lines that were grown hydroponically under +P conditions (200 μM Pi) and −P (5 μM Pi) conditions for an additional 21 d. The Pi concentration in the *OsLPR3*-Ox lines was not significantly different from that in WT in the roots, leaf blades, and leaf sheaths under either +P or −P conditions (Figure 6). However, the Pi concentration of the *OsLPR3* knockout mutant was significantly higher in all tissues than in WT under +P conditions, including in the roots, leaf blades, and leaf sheaths (Figure 6A), and the Pi concentration of the *OsLPR3* knockout mutant was significantly higher in the leaf blades and leaf sheaths than in WT under −P conditions (Figure 6B). Whereafter, we measured the total P concentration in the *OsLPR3* transgene lines and WT. The total P concentrations in the roots, leaf blades, and leaf sheaths of the *oslpr3* mutant were higher in the *oslpr3* knockout lines than in WT (Figure 7A), but they were unchanged under −P conditions (Figure 7B). Furthermore, radioisotope labeling was used to evaluate the effect of *OsLPR3* on the phosphate absorption rate. The uptake of ^32^Pi was significantly higher in the *oslpr3* knockout lines than in the WT and *OsLPR3*-Ox lines under +P conditions but not under −P conditions (Figure 7C,D). The above results indicate that the knockout of *OsLPR3* improved the phosphate utilization capacity of the rice.

### 2.6. Alteration of OsLPR3 Expression Affected the Relative Expressions of OsPTs and Pi-Starvation-Induced Genes

To determine the function of *OsLPR3* in Pi homeostasis, the expressions of the relative genes that maintained Pi homeostasis were explored in the *OsLPR3* knockout lines and WT. The expression levels of the *Pht1* family members were assessed in the roots of the WT and *oslpr3* lines under +P conditions. The transcript levels of *OsPT2*, *OsPT4*, *OsPT8*, and *OsPT10* were significantly higher in the *oslpr3* mutants than in WT, whereas the transcript levels of *OsPT1*, *OsPT6*, and *OsPT9* were not significantly different between the *oslpr3* mutants and WT (Appendix A). Further, the relative expression levels of Pi-starvation-inducible genes, including *OsIPS1*, *OsPAP10a*, and *OsSQD2*, were determined in the roots of the WT and *oslpr3* lines under +P and −P conditions. The relative expression levels of *OsIPS1*, *OsPAP10a*, and *OsSQD2* in the *oslpr3* mutants were not significantly different from those in WT under +P conditions. Under −P conditions, the relative expression levels of *OsIPS1*, *OsPAP10a*, and *OsSQD2* in the *oslpr3* mutants were severely repressed compared with those in WT. The relative expressions of *OsIPS1*, *OsPAP10a*, and *OsSQD2* were highly induced in response to Pi deficiency in WT, which is consistent with a previous study [35]. Notably, the relative expression levels of *OsIPS1*, *OsPAP10a*, and *OsSQD2* in the *oslpr3* mutants were only a little higher under −P conditions than under +P conditions, but the relative expression levels of these genes were far higher in WT under −P conditions than under +P conditions (Figure 8).

## 3. Discussion

As a model plant for monocotyledon species, rice has a larger genome, a longer growth cycle, and a higher biomass than *A. thaliana*, and the mechanism of the growth regulation of rice is more complex and precise than that of *A. thaliana*. In *Arabidopsis*, a low Pi availability relieves the negative regulation of *AtLPR1* by *AtPRD2* in the ER, and it induces AtLPR1 transport to the plasma membrane. The *AtLPR1* activity in the plasma membrane triggers Fe and callose deposition in the apoplast, which causes impaired movement of the SHORT ROOT (AtSHR) protein, interferes with symplastic communication, and is responsible for root meristem differentiation [18,19,24]. In a previous study, *OsLPR3* (Loc_Os01g03630), a rice homolog of *AtLPR1*, was highly induced at the mRNA level by 21 d of Pi starvation, and it rapidly returned to basal transcript levels after 1 h of Pi resupply to the roots [36]. In this study, the strong induction of the relative expression of *OsLPR3* mRNA to low Pi in the leaf blades, leaf sheaths, and roots (Figure 1) is consistent with an earlier study [30]. However, in previous studies, the relative expression of *AtLPR1* was induced by low Pi at the protein level rather than at the mRNA level [13,19]. *AtLPR2* has been found to be abundant in roots and show an increased expression under NH_4_^+^ conditions compared to under NO_3_^−^ conditions [27]. This suggests a different function of LPR in dicotyledon *Arabidopsis* and monocotyledon rice.

Proteins require signal sequences in order to be secreted to the exterior of the cell. Guided by a signal peptide, proteins enter the cytosol through the ER and are finally secreted outside the cell [37]. AtLPR1 mainly functions as a ferroxidase in the cell wall. To be transported to the exterior of the cell, a protein requires a signal peptide to first deliver it to the ER for translocation through the endomembrane system to the plasma membrane. *OsLPR5* and *AtLPR1/2* have a putative ER-specific signal peptide, and they are localized to the ER and cell wall [24,27,38], and *LPR1/2* are required to oxidize Fe(II) and maintain Fe(III)-citrate stability and mobility during xylem translocation against photoreduction [39]. However, *OsLPR3* did not appear to have an ER-specific signal peptide sequence and was different from *OsLPR5* and *AtLPR1/2* (Appendix A). The purified pCold-OsLPR3 fusion protein was heterologously expressed, and the pCold protein and pGS-OsLPR5 were used as negative and positive controls, respectively. The ferroxidase activity of the OsLPR3 fusion protein was comparably lower than that of the pCold negative control and significantly lower than that of the OsLPR5 fusion protein (Appendix A). Furthermore, *OsLPR3* overexpression showed ~20% increased ferroxidase activity under +P conditions but showed ~35% decreased ferroxidase activity under −P conditions (Appendix A). However, in a previous study, *OsLPR5* overexpression showed more than a 430% increase in the ferroxidase activity under both +P and −P conditions [38]. Hence, OsLPR3 might not function as a ferroxidase and, thus, differs from OsLPR5.

Different members of a gene family may be involved in functional conservation or redundancy [40,41,42]. In a previous study, the expressions of *OsLPR3* and *OsLPR5* were significantly induced and repressed, respectively, in *ospho2* mutants [30]. In *OsPDR2* RNAi lines, the expressions of *OsLPR3* and *OsLPR5* were decreased and increased, respectively, in −P split roots compared with +P split roots [43]. These results revealed that *OsLPR3* and *OsLPR5* might function differently in mediating Pi deficiency response networks under −Pi conditions. Subsequent research showed that the *OsLPR5* mutant increased Pi uptake but repressed root growth and affected the translocation of Pi from the root to the shoot under −P conditions at the vegetative stage. Furthermore, the knockout of *OsLPR5* impacted the plant height, seed setting rate, 1000-grain weight, and grain yield per plant at the reproductive stage [38]. However, in this study, the *OsLPR3* mutation increased the ^32^P uptake and total P concentration of rice under +P conditions but had no significant effect on the ^32^P uptake and total P of rice under −P conditions (Figure 7), and the mutation of *OsLPR3* increased the Pi concentration of rice under both +P and −P conditions at the vegetative stage (Figure 6). In addition, the mutation of *OsLPR3* increased the 1000-grain weight and seed length of the rice at the reproductive stage (Figure 4D and Figure 5B). The alteration of *OsLPR3* expression did not affect the total tiller number and effective tiller number of the rice. However, the knockout of *OsLPR3* decreased the seed setting rate of the rice (Appendix A), and this could be the reason why the *oslpr3* mutant had a higher 1000-seed weight compared to the wild type but a similar yield. Furthermore, we conducted a correlation analysis between the expression results and the seed production of the *OsLPR3* transgene lines. The correlation coefficient was −0.5105, indicating a strong negative correlation. These results highlight the different roles of the *LPR* family members *OsLPR3 and OsLPR5* in the growth, development, and P utilization of rice, indicating their different functions in response to Pi deficiency. This can provide some ideas for breeding nutrient-efficient and high-yield varieties.

A low Pi availability restricts plant growth and development. To adapt to low or limited Pi availability, plants usually grow longer roots and increase their root–shoot ratio by increasing their root growth more than their shoot growth [44,45]. Here, the number of rice tillers, an important indicator of Pi nutrient status, was positively correlated with tolerance to low Pi stress [46,47,48]. RSA is sensitive to the Pi status of a growth medium [49]. Improved root growth and RSA are thus important traits to increase P-acquisition efficiency [50,51]. In this study, the phenotypes of the *oslpr3* knockout mutants were not significantly different from those of WT under +P conditions. However, under −P conditions, the growth of the *oslpr3* knockout mutants was significantly promoted. The WT plants did not grow young tillers due to a low Pi, whereas the *oslpr3* mutants grew young tillers under −P conditions (Figure 3B). In addition, the plant height, root length, shoot biomass, root biomass, and number of adventitious roots of the *oslpr3* mutants were increased compared with those of WT under −P conditions. Furthermore, the lower root–shoot ratio of the *oslpr3* knockout mutants under −P conditions indicated that they were less sensitive to a low Pi than the WT plants (Appendix A).

Furthermore, the *oslpr3* mutation increased the total P concentration of the rice under +P conditions, and the ^32^P-isotope-labeling assay further confirmed that the Pi uptake rate of the *oslpr3* mutants was higher than that of WT, which might be because the *OsLPR3* knockout stimulated the expressions of *OsPT2*, *OsPT4*, *OsPT8*, and *OsPT10* in the rice (Appendix A). Under −P conditions, the total P concentration and Pi uptake rate of the *oslpr3* mutants were not significantly different from those of WT. However, the *oslpr3* mutants showed a high tolerance to low Pi stress. The increased soluble Pi content of the *oslpr3* mutant lines might have led to the high tolerance to low Pi stress in the *oslpr3* mutants.

Several genes that play pivotal roles in maintaining Pi homeostasis have been identified in rice [35]. *OsIPS1* has been found to be rapidly induced in the Pi-deprived roots of rice [52], *OsPAP10a* encodes an acid phosphatase [53] and *OsSQD2* is involved in sulfolipid biosynthesis activated by Pi starvation in rice [53,54]. The expression levels of the Pi-starvation-inducible genes in the *oslpr3* mutants were analyzed, and the expressions of all three genes were induced in WT under −P conditions compared with +P conditions. However, the expression levels of *OsIPS1*, *OsPAP10a*, and *OsSQD2* were only slightly higher but not significantly different in the *oslpr3* mutants under −P conditions compared with +P conditions, implying that the *oslpr3* mutants were less sensitive to low Pi, and the mutation of *OsLPR3* increased the tolerance to low Pi stress in the rice. The mutations of *LPR1*/2 result in Fe and callose deposition in the apoplast, subsequently impacting the movement of SHR and interfering with symplastic communication, which is responsible for root meristem differentiation [18,19]. There is a possibility that the alteration of *OsLPR3* expression may influence the expression of its homologous genes, thus affecting Fe and callose deposition in the apoplast. The relative expressions of OsPTs and Pi-starvation-induced genes may also be affected by this process.

## 4. Materials and Methods

### 4.1. Plant Materials and Growth Conditions

Rice (*Oryza sativa* L. ssp. japonica) plants, namely, WT (Nipponbare), *OsLPR3* overexpression lines (Ox5, Ox10, Ox29, and Ox31), and *oslpr3* mutant lines (*oslpr3-1*, *oslpr3-2*, and *oslpr3-3*), with a Nipponbare background, were grown hydroponically in a temperature-controlled growth room (14 h light (30 °C)/10 h dark (22 °C) photoperiod, with a relative humidity of ∼70%). Seed germination and seedling growth conditions were as those previously described [30,55]. The hydroponic medium contained 1. 25 mM NH_4_NO_3_, 0.2 mM NaH_2_PO_4_, 0.4 mM K_2_SO_4_, 1 mM CaCl_2_, 1 mM MgSO_4_, 0.009 mM MnCl_2_, 0.075 mM (NH_4_)_6_Mo_7_O_24_, 0.019 mM H_3_BO_3_, 0.155 mM CuSO_4_, 0.02 mM Fe-EDTA, and 0.152 mM ZnSO_4_, and the pH was adjusted to 5.3. The nutrient medium was replaced every 3 d. The soil for the pot experiments was obtained from an experimental farm at Nanjing Agricultural University. Each pot was filled with 15 kg of air-dried soil supplemented with 40 mg Pi kg^−1^ soil. Standard cultural practices recommended for rice were followed as previously described [56].

### 4.2. qRT-PCR

The total RNA was isolated from the rice samples using the TRIzol reagent (Invitrogen), and it was treated with RNase-free DNase. First-strand cDNA was reverse-transcribed from ~1 µg total RNA using the oligo (dT)_18_ primer (Superscript II^TM^ Reverse Transcriptase, Invitrogen). *OsActin1* (LOC_Os03g50885) was used as an internal control for the qPCR. Each qPCR assay was performed in triplicate using an SYBR green-based master mix (Vazyme) on a *StepOnePlus*™ real-time PCR *system* (Applied Biosystems). The relative expression levels of the genes were computed by using the 2^−ΔCT^ method. The gene-specific primers used in this study are listed in Appendix A.

### 4.3. Transient Expression of OsLPR3 in N. benthamiana Leaves for Subcellular Localization

*Agrobacterium*-mediated transformation was used for the transient co-expression of *35S::eGFP::OsLPR3* and *35S::mCherry::HDEL* in the epidermal leaf cells of *N. benthamiana* as previously described [57]. The *N. benthamiana* leaves were collected 2–3 d after infiltration. Using a diode laser for excitation, EGFP and mCherry fluorescence was visualized at 488 nm and 561 nm, respectively, under a confocal laser scanning microscope (Leica SP8).

### 4.4. Construction of OsLPR3 Overexpression and Mutation Vectors and Generation of Transgenic Plants

For overexpression, the coding sequence (1.605 kb) of *OsLPR3* was amplified from the cDNA isolated from the WT Nipponbare plants using *OsLPR3*-specific primers. The PCR product was then inserted into pCAMBIA1305, which was digested with *Kpn*I and *Bam*HI using a ClonExpress II One Step Cloning Kit (C112-01, Vazyme). For the CRISPR/Cas9-mediated mutation in *OsLPR3*, two gene-specific spacers residing in the exons were selected from the rice-gene-specific spacers library provided by Miao et al., 2013 [58]. The intermediate vector pOs-sgRNA was digested by BsaI (#R0535L, NEB), and then it was ligated to the spacers by using T4 DNA ligase (C301-01, Vazyme). The recombinated vector was then introduced to the final expression vector pH-Ubi-cas9-7 (GATEWAY recombination system, Invitrogen, USA). The constructs were transformed into *Agrobacterium tumefaciens* EHA105 and then into the mature embryos of WT Nipponbare, as described by Upadhyaya et al., 2000 [59].

### 4.5. Southern Blot Analyses

Genomic DNA (~100 μg) was extracted from the leaves of the WT (Nipponbare) and independent overexpression lines. It was then digested with *Eco*RI and *Bam*HI overnight at 37 °C, separated on a 0.8% (*w/v*) agarose gel, and transferred to a Hybond-N^+^ nylon membrane (Amersham). Hybridization was performed with a digoxigenin-labeled hygromycin-resistant gene as the probe at 65 °C overnight. The blots were washed under a stringent condition at 65 °C and analyzed using a phosphorimager (Typhoon-8600) [53].

### 4.6. Quantification of Pi and Total P Concentrations

Pi and total P concentrations were quantified as described by Zhou et al. (2008).

### 4.7. ^32^Pi Uptake Assay

The seedlings (10-d-old) of the WT, *OsLPR3*-Ox, and *oslpr3* lines were grown hydroponically under +P or −P conditions for 7 d. Subsequently, these seedlings were grown for 24 h in a +P and/or −P uptake solution (200 mL) indicated by 8 μCi of ^32^Pi (KH_2_PO_4_, Perkin-Elmer). After the uptake, apoplastic ^32^Pi was removed by incubating the roots of the seedlings in an ice-cold desorption solution (2 mM MES (pH 5.5), 0.5 mM CaCl_2_, 0.1 mM NaH_2_PO_4_) for 10 min. The seedlings were blotted dry, their roots and shoots were harvested separately, and their fresh weights were determined. The plant parts were digested in a mixture containing HClO_4_ and 30% (*v*/*v*) H_2_O_2_ at 28 °C for 8–12 h. A scintillation cocktail (3 mL) was added to the digested tissue, and ^32^Pi activity was determined using a liquid scintillation counter (Tri-Carb 2100, Packard). The uptake of ^32^Pi was calculated by summing the ^32^Pi activity in the root plus shoot and dividing it by the fresh weight of the root for each plant separately.

### 4.8. Statistical Analysis

Data were analyzed for significant differences using IBM SPSS Statistics 20. (http://www-01.ibm.com/software/analytics/spss/, accessed on 3 September 2022).

## Figures and Tables

**Figure 1 ijms-24-02437-f001:**
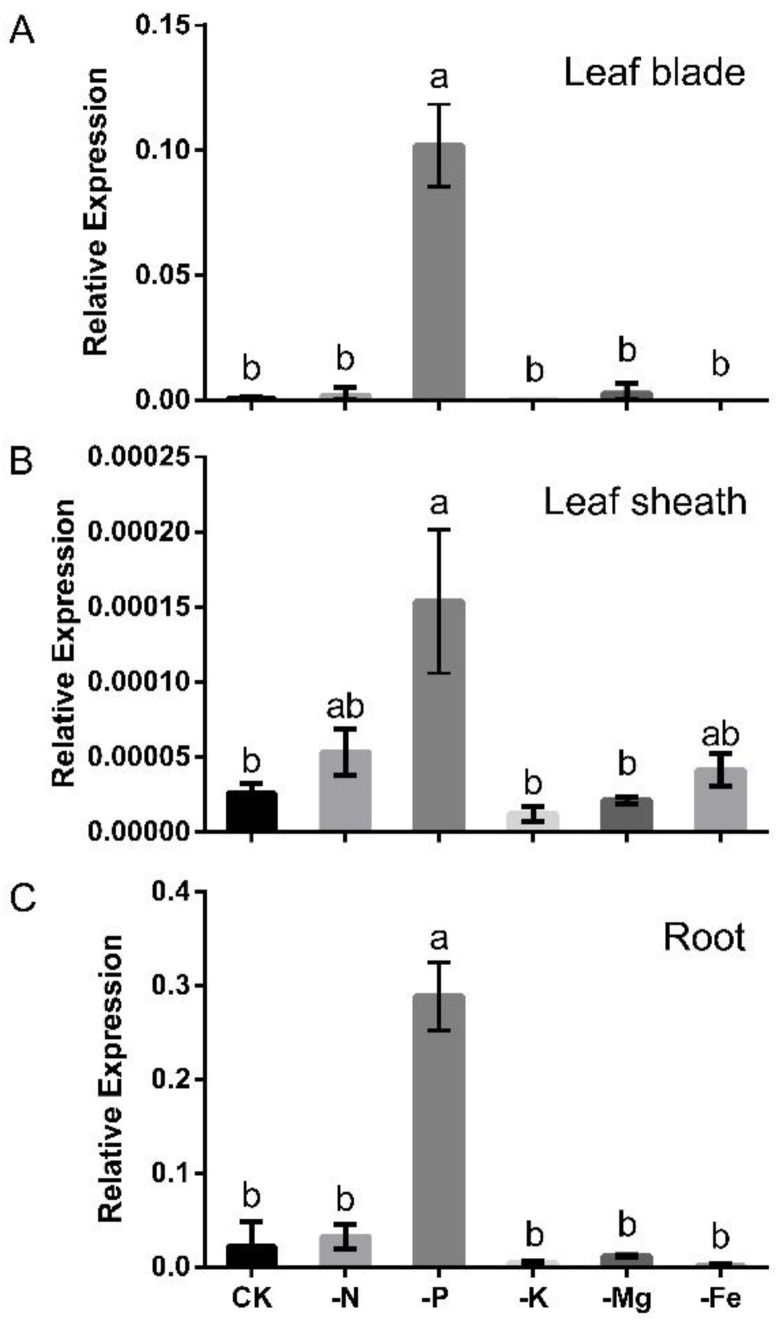
Variable effects of nutrient deficiencies on *OsLPR3* transcript levels. Wild-type rice (Nipponbare) seedlings (10-d-old) were grown for 7 d in complete nutrient solution (CK) or in nutrient solution lacking nitrogen (−N), phosphorus (−P), potassium (−K), magnesium (−Mg), or iron (−Fe). Relative *OsLPR3* transcript levels of leaf blades (**A**), leaf sheaths (**B**), and root (**C**) samples were determined and compared with *OsActin1* via qPCR. Values are means ± SE (*n* = 3). Different letters above the bars indicate significant differences in the relative *OsLPR3* transcript levels (*p* < 0.05, one-way ANOVA).

**Figure 2 ijms-24-02437-f002:**
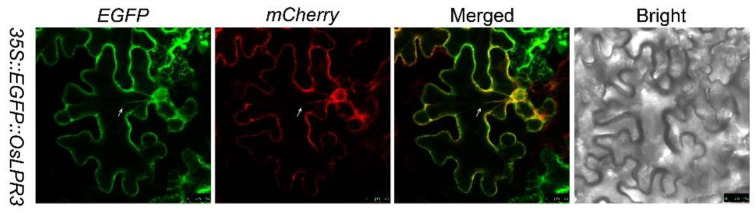
Subcellular localization of OsLPR3. *Nicotiana benthamiana* plants were infected with *Agrobacterium tumafaciens* EHA105 containing the reporter constructs *35S::eGFP::OsLPR3* and *35S::mCherry::HDEL*. Confocal microscopy was used to capture images of epidermal leaf cells showing transient expression of *35S::eGFP::OsLPR3* (EGFP) and *35S::mCherry::HDEL* (CHERRY). The EGFP and mCherry fluorescence images were merged. A bright-field image is included for comparison. Scale bar = 25 μm.

**Figure 3 ijms-24-02437-f003:**
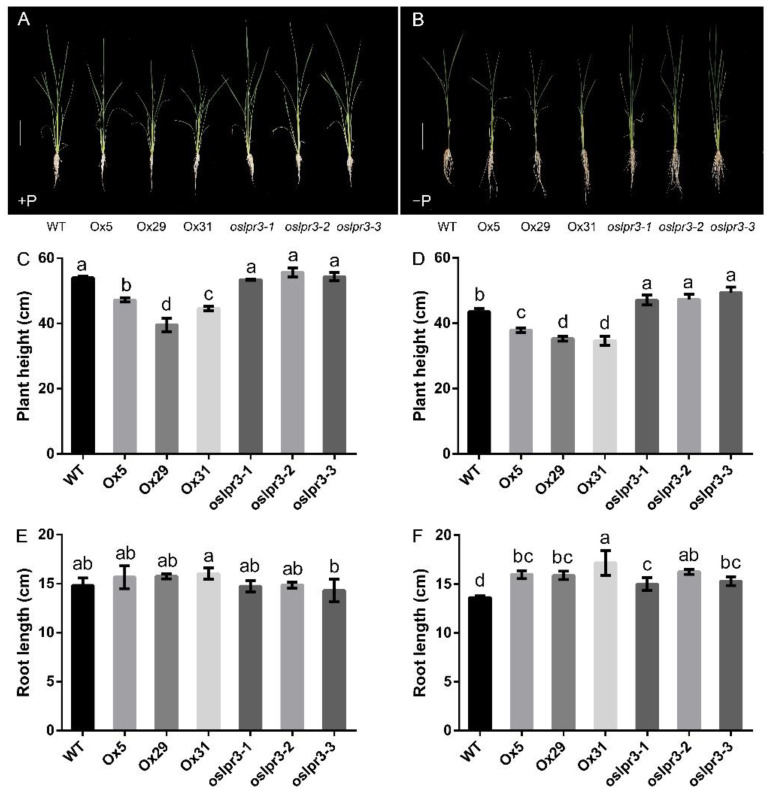
Growth of *OsLPR3*-Ox and *oslpr3* lines in response to Pi supply. Seedlings (10-d-old) of WT, *OsLPR3*-Ox, and *oslpr3* lines were grown hydroponically under +P and −P conditions for 21 d. (**A**,**B**) Photographs of the relative growth performances of WT, *OsLPR3*-Ox, and *oslpr3* lines under different Pi regimes. Scale bar = 10 cm. (**C**,**E**) Plant height and root length under +P conditions and (**D**,**F**) under −P conditions. Values are means ± SE (*n* = 6). Different letters above the bars indicate significant differences (*p* < 0.05, one-way ANOVA).

**Figure 4 ijms-24-02437-f004:**
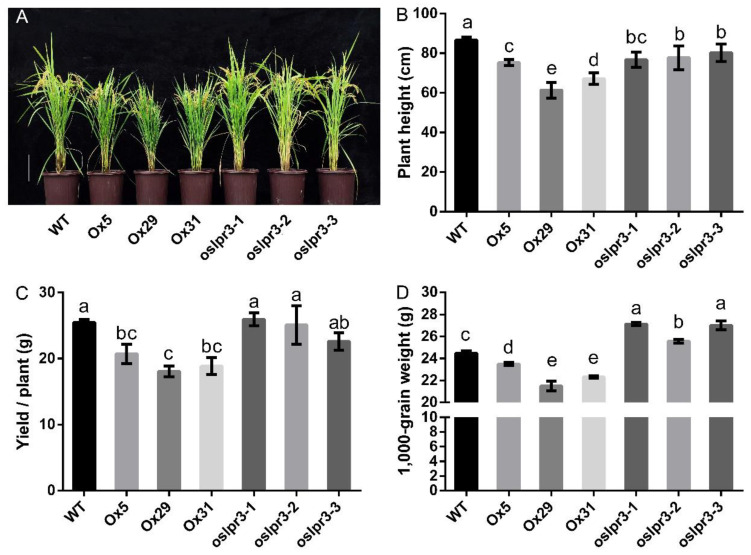
*OsLPR3* affects growth performance and reproductive traits. WT, *OsLPR3*-Ox, and *oslpr3* lines were grown to maturity (20 weeks), (**A**) Photographs of the plant phenotypes. Scale bar = 20 cm. (**B**–**D**) Plant height, plant yield, and 1000-grain weight, respectively. Values are means ± SE (*n* = 10). Different letters above the bars indicate significant differences (*p* < 0.05, one-way ANOVA).

**Figure 5 ijms-24-02437-f005:**
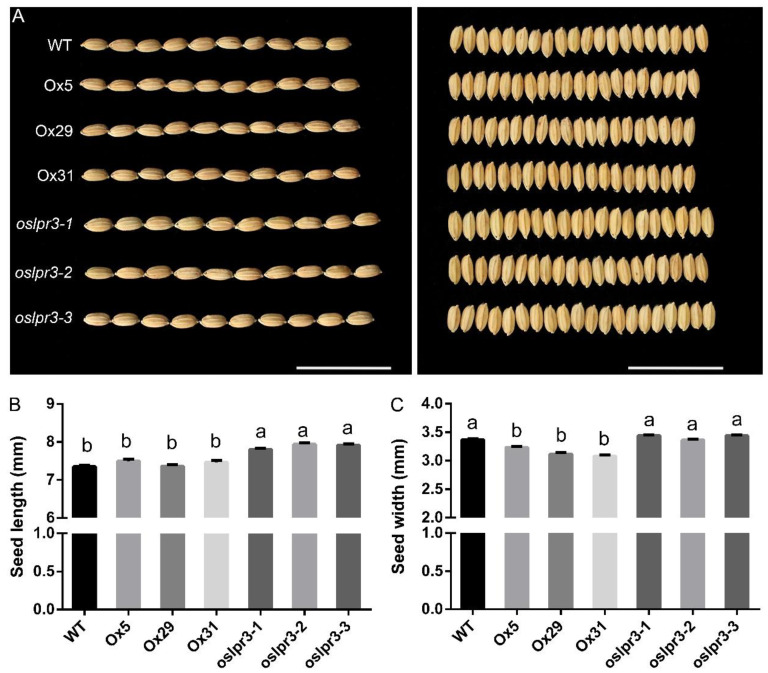
Alteration of *OsLPR3* expression affects seed length and width. WT, *OsLPR3*-Ox, and *oslpr3* lines were grown as described in the legend in Figure 6. (**A**) Photographs of the seed phenotype. Scale bar = 3 cm. (**B**,**C**) Seed length and seed width, respectively. Values are means ± SE (*n* = 30). Different letters above the bars indicate significant differences (*p* < 0.05, one-way ANOVA).

**Figure 6 ijms-24-02437-f006:**
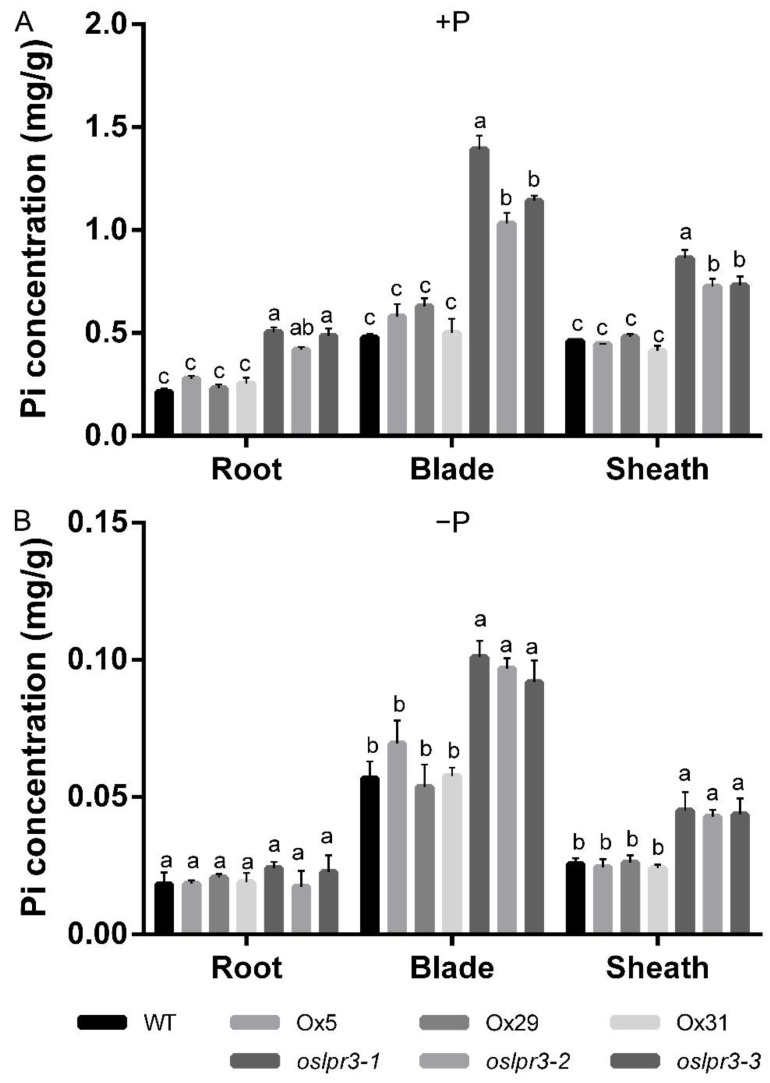
Pi concentration in *OsLPR3*-Ox and *oslpr3* lines. Seedlings (10-d-old) of WT, *OsLPR3*-Ox, and *oslpr3* lines were grown hydroponically under +P (**A**) and −P (**B**) conditions for 21 d. The Pi concentration was determined in roots, leaf blades, and leaf sheaths. Values are means ± SE (*n* = 5). Different letters above the bars indicate significant differences (*p* < 0.05, one-way ANOVA).

**Figure 7 ijms-24-02437-f007:**
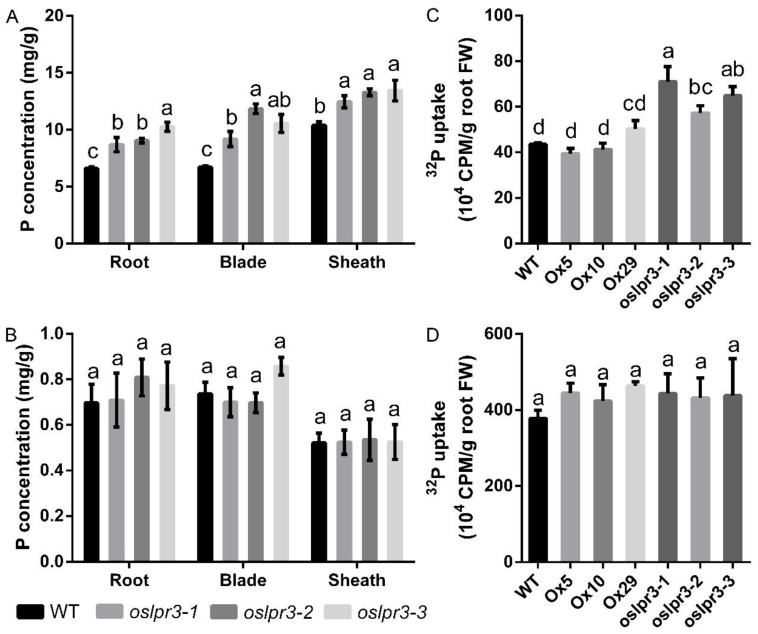
Total P concentration in *oslpr3* lines. (**A**,**B**) Seedlings (10-d-old) of WT and *oslpr3* lines were grown hydroponically under +P (**A**) and −P (**B**) conditions for 21 d. The total P concentration was determined in roots, leaf blades, and leaf sheaths. Values are means ± SE (*n* = 5). (**C**,**D**) Seedlings (10-d-old) of WT, OsLPR3-Ox, and *oslpr3* lines were grown hydroponically under +P (**C**) and −P (**D**) conditions for 7 d. The ^32^Pi uptake over 3 h was determined in WT and OsLPR3 transgenic lines. Values are means ± SE (*n* = 4). Different letters above the bars indicate significant differences (*p* < 0.05, one-way ANOVA).

**Figure 8 ijms-24-02437-f008:**
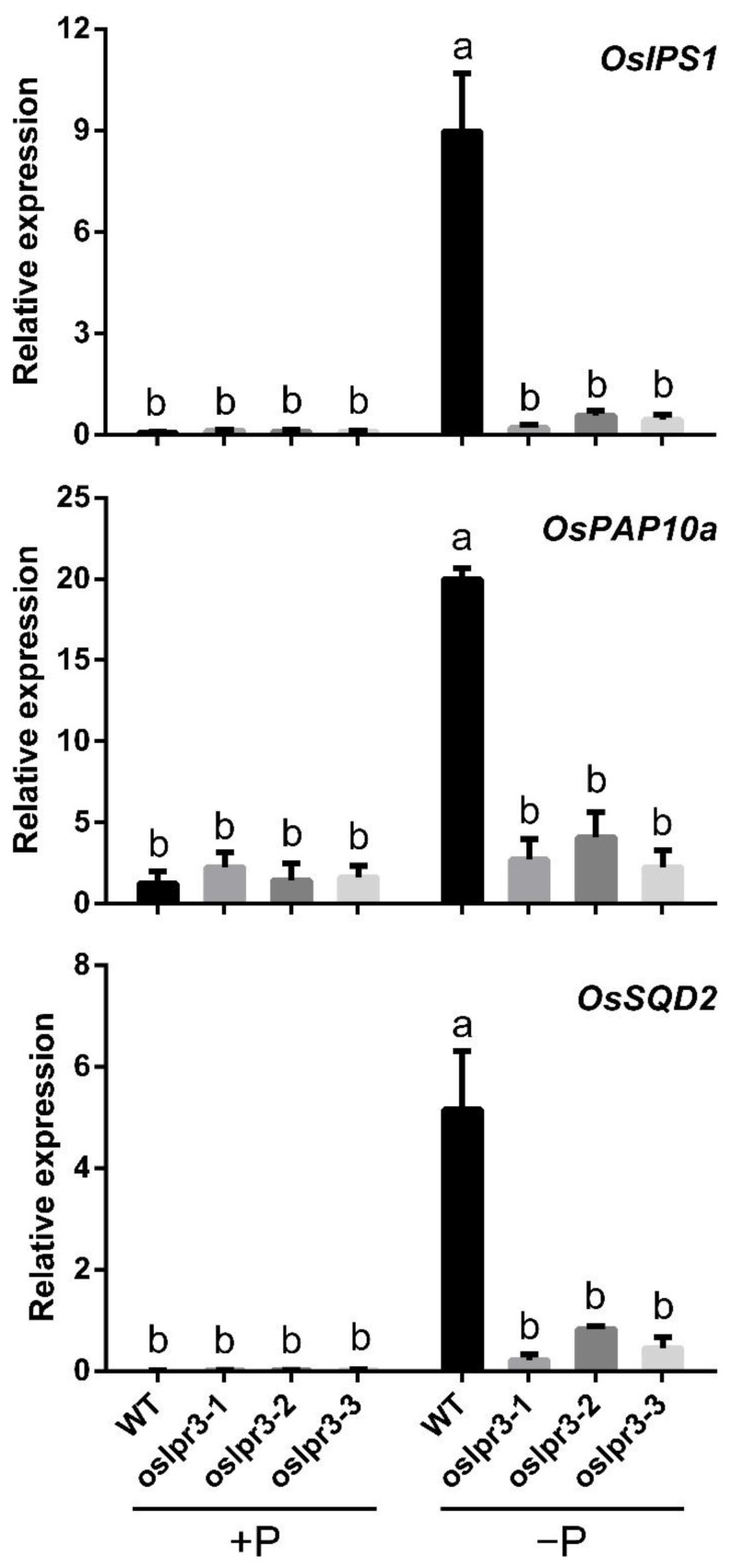
Expressions of Pi-starvation-induced genes in WT and oslpr3 mutants. Seedlings (10-d-old) of WT and *oslpr3* lines were grown hydroponically under +P (200 μM Pi) and −P (5 μM Pi) conditions for 10 d. Relative transcript levels of *OsIPS1*, *OsPAP10a*, and *OsSQD2* in the roots were determined and compared with those of *OsActin* via qPCR. Values are means ± SE (*n* = 3). Different letters above the bars indicate significant differences (*p* < 0.05, one-way ANOVA).

## Data Availability

The data underlying this article are available in the article and in its online Appendix A.

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
