# Peer review of "Mutation of OsLPR3 Enhances Tolerance to Phosphate Starvation in Rice"

_ijms, 2023, doi:10.3390/ijms24032437_

Round 1

Reviewer 1 Report

Manuscript "Mutation in OsLPR3 enhances the tolerance of phosphate starvation in rice" by Ai et al is well research designed and nicely presented. 

CRISPR/Cas9 induced mutation in OsLPR3 gene tends to show better performance compared to wild type under low phosphate. This will have great practical implication.

Few comments

1. Author showed mutant has higher 1000-seed weight compared to wild type but similar yield. I wonder if they notice low number of seeds in mutant. Also in figure 4a, mutant plants seems to have higher tiller number. Do they have tiller number data and how many tillers produce seeds.

2. Data number in figure 1B look odd. Please check for accuracy.

3. Need better explanation in section results 2.5 about figure 6 and figure 7. it is not very clear what data is about in each figure

Reviewer 2 Report

Here the functions of OsLPR3 were explored using transgenic plants in which OsLPR3 was overexpressed (Ox) and knocked out. The authors find oslpr3 knockout mutants were less sensitive to low Pi compared with WT plants and.had improved RSA and Pi utilization, and exhibited higher tolerance to low Pi stress. The overexpressed lines seemed to have a spindly RSA, whereas the oslpr3 lines had a thicker RSA compared with WT. However,why did both Ox and knockout lines have similarly long roots under −P conditions compared with WT?  How to affect the relative expression of OsPTs and Pi starvation induced genes by alteration of OsLPR3 expression? Since five LPR1 homologs in rice, OsLPR3 and OsLPR5 revealed strong tissue-specific induction during Pi deficiency; The authers should add double mutant data to distinguish their functional differences

Author Response

Here the functions of OsLPR3 were explored using transgenic plants in which OsLPR3 was overexpressed (Ox) and knocked out. The authors find oslpr3 knockout mutants were less sensitive to low Pi compared with WT plants and.had improved RSA and Pi utilization, and exhibited higher tolerance to low Pi stress.

Point 1: The overexpressed lines seemed to have a spindly RSA, whereas the oslpr3 lines had a thicker RSA compared with WT. However, why did both Ox and knockout lines have similarly long roots under −P conditions compared with WT?

Response 1: Thanks for your comments.

We totally agree with you and find it a little stange and difficult to explain.

The atlpr1/2 mutant show longer primary root under Pi deficiency mudium (Reymond et al., 2006), there's no problem that OsLPR3 knockout lines have long roots under −P conditions compared with WT. However, the overexpressed lines have similarly long roots under −P conditions compared with WT, this maybe the OsLPR3 overexpression lines were divered by CaMV 35S promoter. Since OsLPR3 mainly express in root, the ectopic expression of OsLPR3 in shoot may result in some difficult problems to interpret.

Point 2:  How to affect the relative expression of OsPTs and Pi starvation induced genes by alteration of OsLPR3 expression?

Response 2: Thanks for your comments.

LPR1/2 were reported participating to low phosphate response, the mutation of LPR1/2 result in Fe and callose deposition in the apoplast., subsequent impact the movement of SHORT ROOT (SHR) and interferes with symplastic communication that is responsible for root meristem differentiation (Müller et al., 2015; Mora-Macías et al., 2017). There is possibility that the alteration of OsLPR3 expression may influenced the expression of its homologous genes, thus affecting Fe and callose deposition in the apoplast. The relative expression of OsPTs and Pi starvation induced genes may also be affected by this process. We have added this content to “Discussion” in our revised manuscript.

Point 3: Since five LPR1 homologs in rice, OsLPR3 and OsLPR5 revealed strong tissue-specific induction during Pi deficiency; The authers should add double mutant data to distinguish their functional differences.

Response 3: Thanks for your comments.

We totally agree with you. OsLPR3 and OsLPR5 revealed similar expression pattern, and may be function partially redundancy. and there are also some functional differentiations between them. To better understand the functional of OsLPR3 and OsLPR5, double mutant is necessary. Actullay, we're doing some research on that, if we get relevant research results, we will report them in future article.

Reviewer 3 Report

The results presented can be considered as a continuation of the analysis of the possible role of LPR genes in the response of rice plants to different phosphate conditions. The authors have performed a number of experiments regarding the possible function of one of the members of the family. The results are not concluding and some of the differences observed in overexpressing or mutant lines are hardly significant. If this is so, the authors could summarize the relation of the expression results with those of results on seed production.

There are no details on the mutations produced by the CRISPR-Cas9 method. Where the same mutations produced in all the independent experiments produced?

Author Response

The results presented can be considered as a continuation of the analysis of the possible role of LPR genes in the response of rice plants to different phosphate conditions.

Point 1: The authors have performed a number of experiments regarding the possible function of one of the members of the family. The results are not concluding and some of the differences observed in overexpressing or mutant lines are hardly significant. If this is so, the authors could summarize the relation of the expression results with those of results on seed production.

Response 1: Thanks for your comments.

We conducted correlation analysis between the expression results and the seed production of OsLPR3 transgene lines as suggest. The correlation coefficient was -0.5105, indicating that they presented a strong negative correlation. This will provide some ideas for breeding high-yield varieties.

We have added this content to “Discussion” in our revised manuscript.

Point 2: There are no details on the mutations produced by the CRISPR-Cas9 method. Where the same mutations produced in all the independent experiments produced?

Response 2: Thanks for your comments.

We have added more details to section “4.4. Construction of OsLPR3 overexpression and mutation vectors, and generation of transgenic plants”. The method of oslpr3 mutations produced by the CRISPR-Cas9 was described as “For CRISPR/Cas9-mediated mutation in OsLPR3, two gene-specific spacers residing in the exons were selected from the rice gene specific spacers library provided by Miao et al., 2013. The intermediate vector pOs-sgRNA was digested by BsaI (NEB #R0535L), and then ligated to the spacers by using T4 DNA ligase (Vazyme C301-01). The recombinated vector was then introduced to the final expression vector pH-Ubi-cas9-7 (GATEWAY recombination system, Invitrogen, USA). The constructs were transformed into Agrobacterium tumefaciens EHA105 and then into the mature embryos of WT Nipponbare as described by Upadhyaya et al. 2000”.

Round 2

Reviewer 2 Report

The authors responded well to my comments, and I agree to accept it.

Reviewer 3 Report

The authors have answered the most relevants questions